# Moderating Effects of BDNF Genetic Variants and Smoking on Cognition in PTSD Veterans

**DOI:** 10.3390/biom11050641

**Published:** 2021-04-26

**Authors:** Gordana Nedic Erjavec, Matea Nikolac Perkovic, Lucija Tudor, Suzana Uzun, Zrnka Kovacic Petrovic, Marcela Konjevod, Marina Sagud, Oliver Kozumplik, Dubravka Svob Strac, Tina Peraica, Ninoslav Mimica, Ana Havelka Mestrovic, Denis Zilic, Nela Pivac

**Affiliations:** 1Laboratory for Molecular Neuropsychiatry, Division of Molecular Medicine, Rudjer Boskovic Institute, 10000 Zagreb, Croatia; gnedic@irb.hr (G.N.E.); mnikolac@irb.hr (M.N.P.); Lucija.Tudor@irb.hr (L.T.); marcela.konjevod@irb.hr (M.K.); dsvob@irb.hr (D.S.S.); 2Department for Biological Psychiatry and Psychogeriatrics, University Psychiatric Hospital Vrapce, 10090 Zagreb, Croatia; suzana.uzun@gmail.com (S.U.); zrnka.kovacic@gmail.com (Z.K.P.); okozumplik@hotmail.com (O.K.); ninoslav.mimica@bolnica-vrapce.hr (N.M.); 3School of Medicine, Josip Juraj Strossmayer University of Osijek, 31000 Osijek, Croatia; 4School of Medicine, The University of Zagreb, 10000 Zagreb, Croatia; MarinaSagud@mail.com; 5Department of Psychiatry, University Hospital Center Zagreb, 10000 Zagreb, Croatia; 6Department of Psychiatry, University Hospital Dubrava, 10000 Zagreb, Croatia; tina.peraica@gmail.com; 7Rochester Institute of Technology Croatia, 10000 Zagreb, Croatia; havelka2@yahoo.com; 8AXON LAB d.o.o., 10000 Zagreb, Croatia; Denis.Zilic@axonlab.com

**Keywords:** PTSD, cognition, BDNF rs6265, BDNF rs56164415, war veterans, smoking

## Abstract

Posttraumatic stress disorder (PTSD) is frequently associated with cognitive disturbances and high prevalence of smoking. This study evaluated cognition in war veterans with PTSD and control subjects, controlled for the effect of smoking and brain derived neurotrophic factor (BDNF) rs6265 and rs56164415 genotypes/alleles. Study included 643 male war veterans with combat related PTSD and 120 healthy controls. Genotyping was done by real time PCR. Cognitive disturbances were evaluated using the Positive and Negative Syndrome Scale (PANSS) cognition subscale and the Rey-Osterrieth Complex Figure (ROCF) test scores. Diagnosis (*p* < 0.001), BDNF rs56164415 (*p* = 0.011) and smoking (*p* = 0.028) were significant predictors of the cognitive decline in subjects with PTSD. BDNF rs56164415 T alleles were more frequently found in subjects with PTSD, smokers and non-smokers, with impaired cognition, i.e., with the higher PANSS cognition subscale scores and with the lower ROCF immediate recall test scores. Presence of one or two BDNF rs56164415 T alleles was related to cognitive decline in PTSD. The T allele carriers with PTSD had advanced cognitive deterioration in smokers and nonsmokers with PTSD, and worse short-term visual memory function. Our findings emphasize the role of the BDNF rs56164415 T allele and smoking in cognitive dysfunction in war veterans with PTSD.

## 1. Introduction

Post-traumatic stress disorder (PTSD) may develop after direct or indirect exposure to a traumatic event [1], with female sex, trauma severity, (family) history of the psychiatric disorder [2], and personality factors such as alexithymia [3] being the most common risk factors for PTSD development. Changes in different neurotransmitter systems, including serotonergic, GABA-ergic, dopaminergic [4] and noradrenergic [5], are found in PTSD. It is a psychiatric disorder in which cognitive functioning has an important role in the development and maintenance (or exacerbation) of symptoms [6]. Impairments of several cognitive systems, including processing speed, learning, memory, and executive function were noted in PTSD [7]. Those alterations could be caused by the structural brain changes reported in PTSD referring to the reduced volumes of the hippocampus and frontal lobe, as well as the total brain volume [8]. It was shown that the presence of re-experiencing and arousal, typical PTSD symptoms, may cause a relative decline in verbal memory [9]. A more recent study by Couette et al. [10] examined social cognition, defined as the ability to perceive, process, and understand social information, and found that it is significantly disturbed in PTSD. War veterans with combat-related PTSD evaluated using the Rey-Osterrieth Complex Figure (ROCF) test showed worse cognitive functions such as visual-spatial perception and short and long-term visual memory function, than veterans who did not develop PTSD [11].

Cognitive deterioration in combat-related PTSD might be affected by the severity of traumatic symptoms, evaluated using the Clinician-Administered PTSD Scale (CAPS), while more severe PTSD symptoms are sometimes accompanied by psychotic features [12,13,14]. Psychotic symptoms develop after PTSD symptoms and usually encompass hallucinations and delusions [14], assessed with the Positive and Negative Syndrome Scale (PANSS). Traumatic symptoms, measured with the total CAPS scores, were significantly and positively correlated with total PANSS scores in war veterans with PTSD [15].

PTSD is often associated not only with cognitive decline but also with a higher prevalence of smoking [16], heavier smoking [17], and greater nicotine dependence [16]. The severity of PTSD symptoms is strongly associated with problematic smoking outcomes [18], supporting the self-medication theory of smoking [19] and regulation of negative affect, shown by the nicotine-modulated emotional information processing [20]. Individuals with PTSD smoked more than twice as much when compared to individuals from the general population, and smokers experienced more negative effects, trauma history, and had a comorbid psychiatric history [21]. Nicotine could enhance aspects of cognitive function, including motor abilities, attention, and memory [22]. On the other hand, healthy smokers had lower cognitive scores and reduced immediate memory and delayed memory scores than non-smokers [23]. In agreement, long-term heavy smoking induced cognitive impairment and cognitive decline [24] and was associated with poor cognitive results in late life [25].

Brain derived neurotrophic factor is a neurotrophin with a crucial role in the survival and differentiation of neuronal populations during development [26]. It is synthesized in glutamatergic neurons [27], astrocytes [28] and microglia [29], initially as a pre-pro-peptide, further cleaved into pro-BDNF which can then be converted to mature BDNF [30]. Both mature and pro-BDNF are secreted as functional forms of BDNF. By participating in the processes of neurogenesis [31], neuronal survival [32], long-term potentiation (LTP) [30], and GABAergic transmission [33]. BDNF regulates neuronal plasticity underlying learning and memory [34]. Any aberration of these processes can cause impairments in cognition and behavior, evident in various pathological conditions [34,35]. Literature data indicate circulating BDNF as a potential biomarker of memory and cognitive function in healthy adult subjects [36], but also in individuals with cognitive decline-related diseases, such as mild cognitive impairment [37], Alzheimer’s disease [38] and Huntington’s disease [39], which were characterized with the decreased levels of serum BDNF. A negative correlation was found between serum BDNF levels and attention in healthy smokers [23]. Additionally, preclinical data show that chronic nicotine administration increases brain derived neurotrophic factor (BDNF) mRNA levels in the dentate gyrus, CA3, and CA1 subfields of the rat hippocampus [40]. Decreased levels of serum BDNF were found in PTSD [41], while a recent meta-analysis revealed higher plasma (not serum) BDNF levels in PTSD subjects [42] compared to controls. BDNF expressed in the limbic system highly moderates fear and stress responses [43]. Besides, BDNF-related neuroplasticity is a major component maintaining hippocampal integrity [44]. BDNF is assumed to affect synaptic plasticity, plastic changes, and memory consolidation, however, its part in cognition is still uncertain [45].

The BDNF gene, located on chromosome 11p13, has numerous polymorphisms with the single nucleotide polymorphisms (SNPs) Val66Met (rs6265) and C270T (rs56164415), proved to affect its activity. The Val66Met polymorphism is a G/A change resulting in a valine (Val) to methionine (Met) substitution at codon 66 of pro-BDNF and it affects BDNF trafficking and activity-dependent release [46]. It was shown that the presence of the A allele corrupts the extracellular level of BDNF by affecting the ability of pro-BDNF to be packed from Golgi apparatus into secretory vesicles and released into synapse [46]. In line with this, healthy individuals with A allele had significant deficiencies in episodic memory, resulting in poorer cognition [46,47]. The association between BDNF rs6265 and PTSD was suggested [48,49,50,51] in case-control studies.

Regarding cognitive skills, a meta-analysis confirmed the association of the BDNF rs6265 and cognition, since GG homozygotes performed better in memory tasks, while A carriers showed improved neurocognitive performances in executive function [52]. The presence of the A allele was associated with greater severity of lifetime and current PTSD symptoms in a large number of European American U.S. military veterans [53]. The A carriers had reduced competence in judging their spatial processing of navigation performance [54], negative memory bias [55], higher re-experiencing symptoms [53], elevated hyperarousal vulnerability, and increased startle scores [56], compared to GG genotype carriers. In our previous study that included a smaller number of war veterans with or without PTSD, we have detected significant association between cognition (visual-spatial perception and short and long-term visual memory function) determined using the Rey-Osterrieth Complex Figure (ROCF) test and BDNF rs6265 in war veterans with PTSD [11]. Namely, A carriers showed poorer short-term visual memory and attention linked with executive functions, compared to GG genotype carriers [11]. In contrast, the study including Chinese subjects with PTSD revealed different associations between BDNF rs6265 genotypes and cognition, suggesting that carriers of one or two A alleles had better vocabulary and digit sign scores, language intelligence scores, operating intelligence scores, and overall intelligence scores, higher number of sorting and lower number of mistakes (random and continuous) in the Wisconsin card sorting test compared to GG carriers [57].

The rs56164415 polymorphism, a C to T substitution in the 5ꞌ untranslated region of BDNF gene may also influence transcription of the BDNF gene, and thereby its expression [58]. The association of BDNF rs56164415 and PTSD was described [59] and not confirmed [51]. The BDNF rs56164415 C allele was associated with better visual cognitive processing through mechanisms probably associated with the volume of the thalamus [60]. Although BDNF rs56164415 was not confirmed to be associated with Alzheimer’s disease (AD) in a meta-analysis [61], it was associated with altered executive function [62] in AD patients, and a reduction in neurocognitive function in T allele carriers [63].

These findings suggest that BDNF rs6265 and BDNF rs56164415, as well as nicotine dependence, may play an important role in cognitive performance in PTSD. However, the role of BDNF rs6265 in cognition [23,52], especially in PTSD [57] is not clear, while data on the association of the BDNF rs56164415 and PTSD and cognition are missing. Therefore, this study aimed to evaluate changes in cognitive function in war veterans with PTSD and control subjects, controlled for the effect of smoking, and hypothesized that the A allele of the BDNF rs6265 and T allele of the BDNF rs56164415 is associated with cognitive decline in war veterans with PTSD.

## 2. Materials and Methods

### Participants

The study included 643 male war veterans with combat-related PTSD, unrelated Caucasian subjects of Croatian origin. The PTSD subjects were sampled consecutively and included in the study if they fulfilled the inclusion/exclusion criteria. The study did not include 68 subjects who refused to participate in the study. Due to the sampling from different institutions and in different time-periods (in the University Hospital Dubrava Zagreb, in the period from 2004 to 2008, and in the Psychiatric Hospital Vrapce, Zagreb, from 2015 to 2017), the diagnosis of current and chronic PTSD was done and confirmed during the initial assessment by the consensus of two trauma-experienced psychiatrists using SCID based on DSM-IV [64] and DSM-5 criteria [65]. The duration of the PTSD was in the range of 10–22 years. The severity of PTSD was assessed using the CAPS [66]. Participants with PTSD were exposed to similar combat related traumatic events during the Homeland war in Croatia. Inclusion criteria were a PTSD diagnosis and combat experience and written informed consent. Exclusion criteria were drug abuse or alcohol dependence within three months prior to admission, schizophrenia, bipolar disorder, adult ADHD, Alzheimer’s disease (according to DSM criteria), and intellectual disability. The study participants were not treated with any psychotropic medication at least 30 days before sampling.

The control group consisted of 120 male age-matched subjects, sampled at the same time as PTSD subjects in both psychiatric hospitals. The study was approved by the corresponding Ethics Committees and was carried out in accordance with the Helsinki declaration (1975), as revised in 1983. All patients have signed informed consent prior to study procedures.

For 639 subjects with available data for smoking, controls and war veterans with PTSD were classified into smokers (subjects smoking ≥ 10 cigarettes per day, i.e., current smokers, N = 411) and non-smokers (i.e., a group of never smokers and former smokers, N = 228).

Cognition was evaluated using the PANSS cognition subscale and consisted of items P2 (conceptual disorganization), N5 (difficulty in abstract thinking), G10 (disorientation), and G11 (poor attention) [67]. Since a smaller part of our war veterans (N = 199) were already assessed using the ROCF test and revealed a significant association of cognition with the BDNF rs6265 [11], in the present study the ROCF testing was used only to evaluate the possible association of cognition with BDNF rs56164415. The ROCF test evaluates visuospatial abilities, attention, visual memory, and processing speed [11]. Within the test, a person needs to replicate the drawing of a complex figure (ROCF copy; max 20 scores), reproduce it from the memory 3 min after observation (ROCF immediate recall; max 20 scores) and 30 min after observation (ROCF delayed recall; max 20 scores).

Genotyping of the BDNF rs6265 and BDNF rs56164415 was described in detail before [68]. Genomic DNA was isolated from peripheral blood using a salting-out method [69]. Genotyping was conducted by a real time PCR method using Applied Biosystems^®^ 7300 Real-Time PCR System apparatus and TaqMan^®^ Genotyping Assays (Applied Biosystems, Foster City, CA, USA) according to the manufacturer’s protocol. Assay IDs were C_11592758_10 for rs6265 and C_89097201_10 for rs56164415. Around 10% of randomly selected samples were genotyped again as quality control for genotyping assays.

Minor allele frequency (MAF) for BDNF rs6265 (A allele) in our sample was 23% in controls and 20% in PTSD, which agrees with the MAF of 20% in the European population [70], while MAF frequency for BDNF rs56164415 (T allele) was 12% for controls and 14% for PTSD%, which is not in line with the much smaller frequency of T allele (6%) in the European population.

In this study we assessed genotype and allelic frequency of both BDNF SNPs, but due to the low number of the BDNF rs6265 AA genotype carriers and BDNF rs56164415 TT genotype carriers in the whole sample, dominant model for the BDNF rs6265: A carriers (AA + AG) vs. GG homozygous genotype, and the dominant model for the BDNF rs56164415: T carriers (TT + TC) vs. CC homozygous genotype [61] was also used.

Statistics software Sigma Stat 3.5 (Jandell Scientific Corp. San Raphael, CA, USA) was used for all evaluation of the results. A General linear model was used to determine the possible effects of diagnosis, smoking, age, and BDNF rs6265 and BDNF rs56164415 polymorphisms on PANSS cognition subscale scores. All cognitive scores deviated from the normal distribution (Kolmogorov-Smirnov test) and were expressed as median, 25th (Q1), and 75th (Q3) percentile. Scores in carriers of different genotype/allelic groups were evaluated with the non-parametric Mann–Whitney test (for two groups) and Kruskal–Wallis ANOVA (for three groups) followed by the Dunn’s test. Haplotype analysis for BDNF rs6265 and BDNF rs56164415 polymorphism, performed using Haploview software v. 4.2, showed that these two polymorphisms were not in linkage disequilibrium (D′ = 0.33), and SNPs were analyzed separately. The χ^2^ test was used for Hardy–Weinberg equilibrium and to calculate differences in BDNF rs6265 and rs56164415 genotypes, alleles, and dominant models. All tests were two-tailed. Since we evaluated two SNPs, the *p* value was reduced to 0.025. G∗Power 3 Software [71] was used to determine *a priori* sample size and statistical power with a = 0.025; expected small to medium effect size = 0.3 for a Mann–Whitney test; 0.20 for a χ^2^ test and 0.15 for Kruskal–Wallis ANOVA; and statistical power (1 − b) = 0.800. The required sample sizes were 426 for a Mann–Whitney 288 for χ^2^ with df = 2 or 238 with df = 1 and 516 for Kruskal–Wallis ANOVA. Therefore, as the study included 784 subjects, it had an adequate sample size and statistical power to detect significant differences among the groups.

## 3. Results

Demographic and clinical data are presented in Table 1. Subjects with PTSD were significantly younger than control subjects (*p* < 0.001) and were more frequently smokers, although this effect was only nominally significant (*p* = 0.031) and did not reach the level of significance after correction. Veterans with PTSD smoked a significantly higher number of cigarettes per day (*p* < 0.001) than healthy control subjects (Table 1). The PANSS scores were determined in both the control and PTSD subjects and, as expected, control subjects had minimal PANSS total scores, while veterans with PTSD had relatively low, although, significantly higher PANSS total scores than control subjects (*p* < 0.001; Table 1). The ROCF scores differed significantly between cases/controls for the ROCF immediate recall and the ROCF delayed recall scores, while ROCF copy recall scores were similar between the groups (Table 1). CAPS scores are presented for veterans with PTSD, and the number and percent of veterans with mild (range 45–65 CAPS scores), moderate (range 66–95 CAPS scores), and severe PTSD (range 96–136 CAPS scores) are presented in Table 1.

### 3.1. The BDNF rs6265 and BDNF rs56164415 Frequency in Veterans with PTSD and Control Subjects

The BDNF rs6265 genotype distribution was in the Hardy–Weinberg equilibrium (HWE) in controls (χ^2^ = 1.606; df = 1; *p* = 0.205) and in veterans with PTSD (χ^2^ = 3.755; df = 1; *p* = 0.054). The distribution of the BDNF rs56164415 genotypes was in the HWE in veterans in PTSD (χ^2^ = 0.321; df = 1; *p* = 0.571), but not in control subjects (χ^2^ = 103.284; df = 1; *p* < 0.001). The deviation of BDNF rs56164415 genotypes from the HWE was confirmed before in the much larger group of control subjects (N = 587) that was not included in this study (χ*^2^* = 103.980; df = 1; *p* < 0.001)

The BDNF rs6265 genotype and allele frequency and BDNF rs56164415 allele frequency were similar between subjects with PTSD and control subjects. The BDNF rs56164415 genotypes were differentially distributed between the two groups (*p* < 0.001). Control subjects were significantly more often carriers of TT genotype compared to subjects with PTSD, while veterans with PTSD were more often CT heterozygotes, compared to control subjects (Table 2). Veterans with PTSD were more often T carriers (TT and CT carriers) than control subjects (Table 2).

### 3.2. The Association of BDNF rs6265 and BDNF rs56164415 Polymorphism with Cognition

A General linear model was used to determine the effects of diagnosis, smoking, age, and two BDNF polymorphisms on the PANSS cognition subscale score. The results (adjusted R^2^ = 0.240) demonstrated that diagnosis (F = 146.406; *p* < 0.001), BDNF rs56164415 (F = 4.524; *p* = 0.011) and smoking (F = 4.821; *p* = 0.028) were significant predictors, while BDNF rs6265 (F = 2.558; *p* = 0.078) and age (F = 0.182; *p* = 0.670) did not contribute significantly to the model.

Cognitive decline in control subjects and veterans with PTSD was assessed with the PANSS cognitive subscale. As expected, control subjects had a minimum score (4 (4; 5)) on the PANSS cognitive subscale, revealing no cognitive disturbances. In contrast, veterans with PTSD showed symptoms of mild cognitive decline, i.e., 6 (5; 8) scores on the PANSS cognitive subscale and these scores were significantly higher (revealing pronounced cognitive decline) than those in control subjects (U = 9270.5; *p* < 0.001).

As revealed with the general linear model, in veterans with PTSD, PANSS cognition subscale scores were significantly associated with BDNF rs56164415 polymorphism (Figure 1), with significant differences found in the PANSS cognitive scores in veterans subdivided into carriers of the rs56164415 genotypes (H = 11.244; df = 2; *p* = 0.004), alleles (U = 87,785.5; df = 2; *p* = 0.008) and T carriers vs. CC homozygotes (U = 33,797.5; df = 2; *p* = 0.002). BDNF rs56164415 TC heterozygotes had the highest scores on the PANSS cognition subscale, and additionally, veterans with PTSD who were carriers of the T allele had increased PANSS cognition scores or stronger cognitive decline than carriers of the C allele (Figure 1).

In veterans with PTSD, PANSS cognitive scores did not differ significantly in subjects subdivided into carriers of the BDNF rs6265 genotypes (H = 5.291; df = 2; *p* = 0.071) or alleles (U = 128,427.0; *p* = 0.426), confirming that BDNF rs6265 did not predict significantly cognitive disturbance measured with the PANSS cognition subscale scores (Figure 1).

In controls, PANSS cognitive scores were similar in subjects subdivided into carriers of the BDNF rs6265 (H = 1.109; df = 2; *p* = 0.574) or BDNF rs56164415 (H = 2.482; df = 2; *p* = 0.289) genotypes, or the BDNF rs6265 (U = 5172.0; *p* = 0.880) or BDNF rs56164415 (U = 2814.0; *p* = 0.049) alleles, respectively (Figure 1).

Since we have recently published the significant association between the BDNFrs6265 and cognition using the ROCF (immediate and delayed recall) test scores [11], in this study, we evaluated only the association between cognition using the ROCF test and BDNF rs56164415. There were no significant differences in the ROCF copy scores in veterans with PTSD subdivided into carriers of the CC, CT, and TT genotypes (H = 0.270; df = 2; *p* = 0.874). However, the ROCF immediate recall scores differed significantly (H = 9.117; df = 2; *p* = 0.010), as carriers of the CT genotype had significantly lower (*p* = 0.008) ROCF immediate recall scores (i.e., the pronounced cognitive deterioration) than carriers of the CC genotype. This result was confirmed as T carriers had significantly decreased ROCF immediate recall scores (U = 1902.0; *p* = 0.008) compared to CC genotype carriers. The ROCF delayed recall scores did not differ significantly when veterans were subdivided into BDNF rs56164415 genotype carriers (H = 3.500; df = 2; *p* = 0.174) or between the T carriers vs CC homozygotes (U = 2265.5; *p* = 0.182).

### 3.3. The Association of BDNF rs6265 and BDNF rs56164415 Polymorphism and Smoking on Cognition

The association of BDNF rs6265 and BDNF rs56164415 polymorphism and smoking was shown in Table 3. Smokers with PTSD had significantly higher total PANSS and higher PANSS cognitive scores than non-smokers with PTSD (Table 3). Control subjects had minimal, similar scores on PANSS total and PANSS cognitive subscale, independent of the smoking status (Table 3). The ROCF copy, immediate recall, and delayed recall scores did not differ significantly between smokers and non-smokers in control subjects as well as in veterans with PTSD (Table 3).

The frequencies of the BDNF rs6265 and BDNF rs56164415 genotypes or alleles did not differ significantly in control and PTSD smokers and non-smokers (Appendix A). A nominally significant trend (*p* = 0.039), was observed in the control group since smokers were more often carriers of the BDNF rs56164415 T allele (16.4%), compared to non-smokers (7.5%), but this trend did not remain significant after correction for multiple testing (Appendix A).

Due to differences in the cognitive scores between smokers and non-smokers, subjects were subdivided depending on the diagnosis and smoking status. Associations of BDNF rs6265 and BDNF rs56164415 polymorphisms with the PANSS cognitive subscale scores were determined in smokers and nonsmokers (Figure 2, Appendix A).

In control subjects who were smokers, PANSS cognitive subscale scores differed significantly when subjects were subdivided according to the BDNF rs6265 genotypes (H = 8.373; df = 2; *p* = 0.015), but analysis of the dominant (U = 493.0; *p* = 0.361) and allelic (U = 1420.5; *p* = 0.813) model did not confirm this association. In control non-smokers, BDNF rs6265 was not associated with PANSS cognitive scale (Figure 2, Appendix A), since cognitive scores did not differ significantly between the carriers of the BDNF rs6265 genotypes (H = 2.010; df = 2; *p* = 0.366), A carriers vs. GG genotype carriers (U = 320.5; *p* = 0.251) or G alleles (U = 1045.0; *p* = 0.462).

In smokers with PTSD, nominally different PANSS cognition scores were found when veterans were subdivided into BDNF rs6265 genotypes (H = 6.647; df = 2; *p* = 0.036) and A carriers (U = 12,090.5; *p* = 0.029), however, allelic model (U = 35,181.5; *p* = 0.104) was not significant (Figure 2, Appendix A). In non-smokers with PTSD, BDNF rs6265 genotypes (H = 0.869; df = 2; *p* = 0.647), dominant (U = 3262.0; *p* = 0.495) or allelic (U = 8681.5; *p* = 0.424) models were not significantly associated with the PANSS cognitive subscale scores (Figure 2, Appendix A).

However, BDNF rs56164415 was significantly associated with cognitive decline in all tested models in veterans with PTSD (both smokers and non-smokers) and in control subjects who were non-smokers. Heterozygotes had the highest PANSS cognitive scores compared to homozygotes for both alleles. Additionally, T carriers had greater cognitive decline, compared to CC homozygotes and C allele carriers (Figure 2, Appendix A).

In control smokers, PANSS cognition scores did not differ between carriers of the BDNF rs56164415 genotypes (H = 0.318; df = 2; *p* = 0.853), T vs. CC carriers (U = 316.5; *p* = 0.695) or T vs. C allele (U = 1194.0; *p* = 0.685) carriers. Opposed to these results, in control subjects who were non-smokers, significantly different PANSS cognitive scores were found between the BDNF rs56164415 genotype (U = 53.0; *p* = 0.001), T vs. CC (U = 53.0; *p* = 0.001) or T vs. C allele (U = 212.0; *p* < 0.001) carriers.

In veterans with PTSD who smoked, BDNF rs56164415 was associated with cognitive decline in smokers in the genotype (H = 10.729; df = 2; *p* = 0.005), dominant (U = 9327.0; *p* = 0.003), and allelic (U = 24113.5; *p* = 0.011) model (Figure 2, Appendix A). In non-smokers with PTSD, the TT homozygous carriers (H = 8.418; df = 2; *p* = 0.015), T carriers (U = 1892.5; *p* = 0.006), and T allele carriers (U = 4614.5; *p* = 0.004) had the highest PANSS cognitive scores compared to scores found in carriers of one or two C allele (Figure 2, Appendix A).

Since smoking status significantly affected the association between BDNF rs56164415 and cognition measured with the PANSS cognition scores in veterans with PTSD, we evaluated this association also with the ROCF test. There were no differences in ROCF Copy, Immediate and Delayed Recall test scores between smokers and non-smokers with PTSD (Table 3).

When we evaluated the possible association between the ROCF copy, immediate and delayed recall test scores and BDNF rs56164415 in all veterans with PTSD, no significant differences were found in the ROCF copy test scores between veterans subdivided into genotypes (H = 1.036; df = 2; *p* = 0.596), T vs. CC carriers (U = 2376.0; *p* = 0.478) and allele carriers (U = 6217.0; *p* = 0.310). Significantly different ROCF immediate recall test scores were detected between BDNF rs56164415 genotype carriers (H = 11.065; df = 2; *p* = 0.004), T vs. CC genotype carriers (U = 1698.0; *p* = 0.003) or T and C allele carriers (U = 5061.0; *p* = 0.014). The scores of the ROCF delayed recall test did not differ significantly in veterans with PTSD subdivided into carriers of the different genotypes (H = 5252.0; df = 2; *p* = 0.072), T vs. CC genotype carriers (U = 2001.0; *p* = 0.071) or T and C allele carriers (U = 5750.0; *p* = 0.185).

When veterans with PTSD were subdivided according to smoking status, the ROCF copy test scores did not differ significantly in smokers with PTSD subdivided into BDNF rs56164415 genotype carriers (H = 0.306; df = 2; *p* = 0.858), T vs. CC genotype carriers (U = 175.5; *p* = 0.771) or T and C allele carriers (U = 433.0; *p* = 0.886). In line with this, in non-smokers with PTSD, ROCF copy test scores did not differ significantly between BDNF rs56164415 genotype carriers (H = 0.410; df = 2; *p* = 0.815), T vs. CC genotype carriers (U = 42.0; *p* = 0.522) or T and C allele carriers (U = 116.0; *p* = 0.464).

The ROCF immediate recall scores were significantly lower (revealing stronger cognitive decline) in veterans with PTSD than in controls (Table 1). In addition, in all veterans with PTSD, heterozygotes (CT carriers) had the lowest ROCFT immediate recall test scores (*p* = 0.004), and the T carriers had significantly lower test scores compared to CC homozygotes (*p* = 0.003) and C allele carriers (*p* = 0.014). This was confirmed in PTSD smokers, as CT genotype carriers had nominally lower ROCF immediate recall scores (H = 7.275; df = 2; *p* = 0.026) than carriers of the other BDNF rs56164415 genotypes, and this relationship was confirmed with significantly decreased ROCF immediate recall scores in T compared to CC genotype carriers (U = 86.0; *p* = 0.024), while there was a trend in T vs. C allele carriers (U = 267.0; *p* = 0.063). In contrast, in non-smokers with PTSD, there were no significant differences in ROCF immediate recall scores between carriers of different BDNF rs56164415 genotypes (H = 0.989; df = 2; *p* = 0.610), T vs. CC genotype carriers (U = 33.5; *p* = 0.390) and T and C alleles (U = 85.0; *p* = 0.232).

In smokers and non-smokers with PTSD, no significant differences in the ROCF delayed recall test scores between carriers of BDNF rs56164 genotypes (H = 2.124; df = 2; *p* = 0.346 for smokers and H = 2.806; df = 2; *p* = 0.246 for non-smokers), T vs. CC genotype carriers (U = 123.5; *p* = 0.167 for smokers and U = 43.0; *p* = 0.766 for non-smokers), and alleles (U = 321.0; *p* = 0.199 for smokers and U = 118.0; *p* = 0.721 for non-smokers) was detected. Detailed results are presented in Appendix A.

## 4. Discussion

To the best of our knowledge, this is the first study to reveal a significant association between BDNF rs56164415 polymorphism with cognitive decline in veterans with PTSD. The results from this study showed that the presence of one or two T alleles of the BDNF rs56164415 was related to cognitive decline in PTSD. This was shown in the higher PANSS cognition scores (showing advanced cognitive deterioration), which was detected both in smokers and in nonsmokers, but also in the lower ROCF immediate recall scores, suggesting worse short-term visual memory function. This is also the first study to show significant effects of smoking on the BDNF rs56164415 polymorphism and cognition in veterans with PTSD.

In line with our previous data [11,72], in this study veterans with PTSD had worse cognitive deterioration than control population, despite being 9 years younger. This cognitive decline in veterans with PTSD was confirmed when cognition was evaluated using the PANSS cognition scores that were significantly higher than in the controls and with the ROCF immediate and delayed recall scores that were significantly lower than in controls. These results are in line with the cognitive decline reported in patients with PTSD who performed poorer than healthy individuals in diverse ranges of cognitive domains, such as executive functioning [73], speed [74,75], and attention [11,74], although this cognitive decline was determined with other cognitive scales.

Although PTSD symptom onset and aging can alter BMI trajectories over time, there was no significant difference in BMI between veterans with PTSD and healthy control subjects. This lack of difference in BMI (and BMI categories) was also confirmed in our previous study including a smaller number of veterans with PTSD, veterans without PTSD, and a large number (more than 1500) of healthy control population-based sample [76]. In addition, BMI values did not differ significantly between 316 veterans with PTSD, subdivided into carriers of the BDNF rs6265 A vs. homozygous GG genotype carriers, or between the BDNF rs56164415 T vs. CC genotype carriers [68].

Smoking status and diagnosis (PTSD) were significant predictors of cognitive alterations. The effect of smoking was significant since smokers with PTSD had higher PANSS total and PANSS cognition scores (i.e., greater cognitive decline) compared to nonsmokers. Our results are in line with reports showing that smoking reduced cognitive scores and immediate and delayed memory scores [23], while long-term heavy smoking elicited cognitive disturbances and cognitive deterioration [24] and weak cognitive performance in older age [25]. Veterans with PTSD smoked more frequently and smoked a higher number of cigarettes per day than control subjects. These results agree with increased smoking and heavier smoking in patients with PTSD [21] since moderate to high nicotine dependence was associated with greater symptomatology of both PTSD and depressive symptoms in another sample of Croatian war veterans [77]. No significant effect of the smoking status on cognition was found in controls using the PANSS and the ROCF test scores. In addition, smoking did not significantly affect cognition in veterans with PTSD evaluated using the ROCF test scores. Therefore, smoking affected differently cognition in controls and veterans with PTSD, depending on the test used.

The distribution of the BDNF rs6265 genotypes, alleles, and the A carriers vs. GG homozygotes was similar between veterans with PTSD and control subjects. This agrees with previous data obtained in Croatian veterans with or without PTSD [11,50] and with the other published data [51,59]. Opposed to our findings, soldiers deployed from Iraq and Afghanistan with probable PTSD had more than three folds higher frequency of the BDNF rs6265 AA genotypes, and two folds higher frequency of A allele carriers than those without probable PTSD [48]. The discrepancy might arise from different ethnicity, given that the higher percentage of AA genotypes in this study population [48]. Moreover, a meta-analysis did not support an association between the BDNF rs6265 polymorphism and PTSD [78]. This polymorphism may not confer risk for PTSD per se, but, nevertheless, may modulate a range of disease features, such as psychotic features [50], or cognition [11]. present study.

In our previous study, the association between the BDNF rs6265 and cognition in veterans with PTSD was detected, showing a significant cognitive decline, evaluated using the ROCF test scores, in carriers of the A allele compared to G allele carriers [11]. In the present study, which included a much bigger sample, but a different scale measuring cognition (i.e., PANSS cognition subscale), the general linear model revealed that BDNF rs6265 did not significantly predict cognitive decline measured with the PANSS cognition scores. These results were confirmed with similar PANSS cognition scores in both control subjects and veterans with PTSD, subdivided into carriers of the different BDNF rs6265 genotypes and alleles. These inconsistencies with our previous study might be explained by the differences in the sample sizes (199 vs. 643 veterans), differences in cognitive tests (ROCF vs. PANSS cognition subscale), and different cognitive domains that are evaluated in our present and previous [11] study. However, in Chinese subjects with PTSD, BDNF rs6265, and the AA genotype, was significantly associated with worse general cognitive function, especially in the executive function, such as generalization, attention, working memory, cognitive transfer, visual discrimination, space perception, planning ability, compared to GG genotype [57].

Smoking significantly affected cognition, and in line with that, in veterans with PTSD who smoked, the BDNF rs6265 A carriers had the highest scores on the PANSS cognitive subscale (representing worse cognition) than GG carriers. In line with these findings, our previous study [11] found poorer performance on the ROCF test in Croatian veterans with PTSD, carriers of the A allele compared to GG genotype carriers. In Korean civilian women with PTSD, the A allele carriers had inferior immediate memory performance than controls, and negative memory bias significantly increased with the increasing number of the A alleles [55]. The presence of the A allele may be associated with the deterioration of some aspects of cognition, as well as other indicators of the poorer outcome in PTSD patients, such as impaired fear extinction [79], development of psychotic symptoms [50], greater skin conductance response to threat [80], greater cortisol suppression [80], greater severity of lifetime and current PTSD symptoms, specifically re-experiencing symptoms [53], lower visuospatial abilities [11], more severe cognitive impairment [57], and greater memory bias [55], compared to GG homozygosity status. Among the military population who served in Iraq or Afghanistan, those with current suicidal ideation were more frequently A allele carriers [81]. Although previous studies did not control for the smoking status [11,50,55,79,80,81], in line with these data, our findings suggest lower cognitive abilities in the A allele carriers in veterans with PTSD who are current smokers. This agrees with the greater impairment of the fear extinction, measured as skin conduction level, related to higher severity of PTSD symptoms, in A compared to GG carriers [79]. Therefore, the presence of A allele and worse cognition in smokers with PTSD might arise from the more severe clinical symptoms. Since the GG homozygosity in healthy individuals increased the negative impact of environmental adversity on hippocampal volume, while A carrier status had the opposite effects [82], these findings suggest that the impact of the A allele is not always simple, and might be explained by the compensatory mechanism [56], or by the fact that BDNF overactivity may also be harmful [83]. BDNF rs6265 has pleiotropic effects in various psychiatric disorders and is associated with multiple phenotypes, and therefore, either A or G allele might exert beneficial or damaging effects [83]. Collectively, studies reveal complex involvement of the BDNF rs6265 on psychological and biological consequences of stress. While inconsistencies in BDNF rs6265 findings related to cognition were suggested to be attributed to ethnicity [84], suicidality [85], age, sex, environmental factors (i.e., prenatal adversities, childhood trauma, life stress), and gene-gene interaction [83], our results were controlled for the effects of age, sex and ethnicity, and added smoking status as a confounding variable.

To our knowledge, only two studies have addressed the association of the BDNF rs56164415 polymorphism and PTSD, while none investigated its relationship with cognition in PTSD. Our results agree with the findings of the increased T allele frequency in individuals with PTSD compared to controls [59] and disagree with the similar frequency of BDNF rs56164415 genotypes in European American patients with PTSD and healthy subjects [51]. Previous studies included male and female individuals with PTSD and recruited Chinese participants with sporadic PTSD [59]. In addition, in contrast to our study, a small sample size, with only 96 PTSD patients, might have affected previously published negative results [51].

In the present study, among veterans with PTSD, BDNF rs56164415 T carriers had worse cognitive scores, compared to C allele or CC genotype carriers. In addition, CT carriers had higher PANSS cognition scores (i.e., greater cognitive decline) than CC carriers. This finding might be interpreted as a positive molecular heterosis effect, where carriers of the heterozygous genotype show a greater effect than homozygous genotype carriers [83]. The significant association between the presence of the T allele in veterans with PTSD and the highest PANSS total and PANSS cognition subscale scores or a greater cognitive decline than in CC genotype carriers partly agrees with results showing that Chinese patients with schizophrenia with the CT genotype had higher PANSS positive scores than those with the CC genotype [86].

To confirm these findings, the association of the BDNF rs56164415 was assessed also using the ROCF test scores. As expected [11], the ROCF copy scores did not differ between veterans with PTSD subdivided into carriers of the CC, CT, and TT genotypes or C and T alleles. BDNF rs56164415 was not associated with the ROCF delayed recall scores. On the other hand, significant differences were found in the ROCF immediate recall scores between T allele carriers and CC genotype carriers. Namely, significantly lower ROCF immediate recall scores were detected in veterans with PTSD, carriers of the CT genotype compared to carriers of the CC genotype, and in T carriers compared to CC genotype carriers. These results show that the T allele presence was associated with worse visual short-term memory and visual object manipulation after few seconds, suggesting a poorer executive function in veterans with PTSD. These findings suggest that the BDNF rs56164415 T allele might represent a possible “risk” allele in cognition assessed in PTSD.

BDNF rs56164415 was reported to be associated with cognition in neurodegenerative disorders since the T allele was more frequently found in patients with Alzheimer’s disease (AD), characterized with significant cognitive impairment [87]. This was confirmed in a German sample [88], in different ethnic cohorts [89,90,91] and in a meta-analysis, showing that the BDNF rs56164415 polymorphism increased AD risk by 88%, but only in the Asian population and only under the dominant (TT/TC vs. CC) model [92]. Another study and a meta-analysis did not replicate this association [61,93], but BDNF rs56164415 was related to altered executive function [62], and the presence of the T allele was associated with reduced neurocognitive function [63] in AD. On the other hand, in a Japanese AD sample, patients with CC homozygosity had better performance on the Frontal Assessment Battery tests (representing executive, but not memory functions), compared to those with CT genotype [62]. However, this polymorphism was not associated with cognition evaluated using the Mini-Mental State Examination (MMSE) scores in AD patients from Turkey [91]. While BDNF rs56164415 polymorphism in predominantly European-Americans patients with AD was not associated with the AD progression, prior educational attainment, or performance on cognitive tests including the MMSE, the T allele carriers had a higher prevalence of neuropsychiatric symptom scores than CC homozygotes, which specifically increased the risk for hallucinations [94]. BDNF rs56164415 polymorphism was also associated with other neurodegenerative disorders, such as amyotrophic lateral sclerosis, where the frequency of the CT genotype and T allele was higher in the patient group of the Han Chinese origin than in controls [95]. These findings suggest the involvement of the BDNF rs56164415 in cognitive processes.

Smoking status significantly affected cognitive performance in controls and in veterans with PTSD, and the presence of the T allele of the BDNF rs56164415 was repeatedly associated with stronger cognitive impairment in both smokers and non-smokers with PTSD. Namely, T allele carriers had significantly higher PANSS cognitive subscale scores (revealing greater cognitive disturbances) than carriers of other genotypes/alleles. Even in control subjects who were non-smokers, TT genotype carriers and T allele carriers had significantly higher PANSS cognitive scores than CC genotype carriers. This was confirmed with the significantly lower ROCF immediate recall scores (suggesting stronger cognitive decline) in all veterans with PTSD, and in smokers with PTSD, who had either one or two T alleles compared to other genotype/allele carriers. There are no data on the relationship between BDNF rs56164415 and cognition, controlled for the effect of smoking, in PTSD.

The mechanism by which BDNF rs56164415 polymorphism may modulate cognitive dysfunction in PTSD is at present unknown. This polymorphism is located in the 5′-non-coding region of the BDNF gene, acting as a functional promoter polymorphism [95], which may result in the altered translation efficacy [87]. It was reported that the C/T substitution might contribute to the loss of transcription factors Histone Nuclear Factor P and ZIC3 binding sites, which was hypothesized to change the efficacy of BDNF translation in the somatic, dendritic, or axonal regions of neurons, producing BDNF imbalances in the cortex [95]. Defective BDNF expression was found in the prefrontal cortex and hippocampus in rats with PTSD-like behaviors induced by traumatic stress [96]. In addition, Vietnam war veterans had higher DNA methylation of the BDNF promoter than veterans without PTSD [97]. Theoretically, under traumatic stress condition, carriers of the T allele may have more pronounced impairment of the BDNF transcription, which can make them more vulnerable to develop cognitive dysfunction in PTSD.

The T allele of the BDNF rs56164415 was slightly more prevalent in healthy smokers, while no such difference was observed in the PTSD group. These data should be viewed in the context of increased T allele frequency, smoking prevalence, and psychopathology in PTSD patients. Namely, veterans with PTSD may smoke to alleviate symptoms, and therefore, smoking might be more related to disease severity than to genetic components. In turn, in the absence of psychopathology as a confounder, genetic factors might be more important in the control group. To the best of our knowledge, there is no data on the BDNF rs56164415 and smoking. While the T allele is a ”risk allele” for neurodegeneration, it may also be a ”risk allele” for smoking, at least in healthy individuals. Given that smoking increases dementia risk [98], BDNF rs56164415 polymorphism may have moderating or additive effects, which deserves future studies.

### Limitations and Strengths

The limitation is that cognition was evaluated with the PANSS cognitive domain, and therefore, this evaluation does not cover neuropsychological testing. Unlike cognitive testing, PANSS derived cognitive subscale is based on the clinical ratings. Therefore, for the significant results, we also added findings from the ROCF test scores, measuring visual-spatial perception as well as short-term and long-term visual memory function. Data on the physical activity were not collected, and the differences in exercising between groups cannot be excluded. Exercise training has many benefits on neuroplasticity and cognition across diagnostic boundaries [99], and, more specifically, greater exercising may modulate the effect of BDNF rs6265 A allele on the severity of PTSD symptoms [53]. Other BDNF gene variants, such as rs908867 and rs925946 in depression [100], and rs7103411, rs988748, and rs7130131 associated with long-term visual memory in healthy controls [101], might also affect cognition and were not assessed. PTSD [102], as well as smoking [103] prevalence, differed among different age groups. Our control subjects were significantly older than PTSD subjects, but general linear regression showed that in our sample cognition was not affected by age.

The strengths of the study are in evaluation of cognition using both PANSS cognition subscale scores and the ROCF test scores, studying of two BDNF polymorphisms, fairly large sample size (N = 784) and adequate statistical power, introducing smoking as a confounder, and being the first study to investigate the association of the BDNF rs56164415 with cognition in PTSD. The sample included was an ethnically homogenous male population, which is important to exclude the possible effect of female sex due to the sexual dimorphism for BDNF rs6265 [83] and to exclude ethnic differences [84]. In addition, PTSD veterans had similar combat-related experiences. As suggested [83], the results were grouped and presented as genotypic (AA vs. GA vs. GG, or TT vs. CT vs. CC), dominant (A carriers vs. GG, or T carriers vs. CC), or allelic (A vs. G, and C vs. T) models.

## 5. Conclusions

This study investigated the role of two common BDNF variants (Val66Met or s6265, and C270T or rs56164415) in Croatian male veterans and controls, on cognitive functions in PTSD, assessed by PANSS cognitive subscale scores, and conformation was done using the ROCF copy, immediate and delayed recall scores. Our findings emphasize the role of the BDNF rs56164415 T allele and smoking in cognitive dysfunction in veterans with PTSD. Given that Croatian veterans with PTSD are now middle-aged men, and the association between PTSD and all-cause dementia, the assessment of BDNF polymorphisms may have a role in establishing which patients may be at particular risk for cognitive deterioration. Our results also implicate the urgency for smoking cessation in this vulnerable population. Finally, both BDNF polymorphisms need to be examined in different neurocognitive tests, in order to elucidate their association with more specific cognitive domains.

## Figures and Tables

**Figure 1 biomolecules-11-00641-f001:**
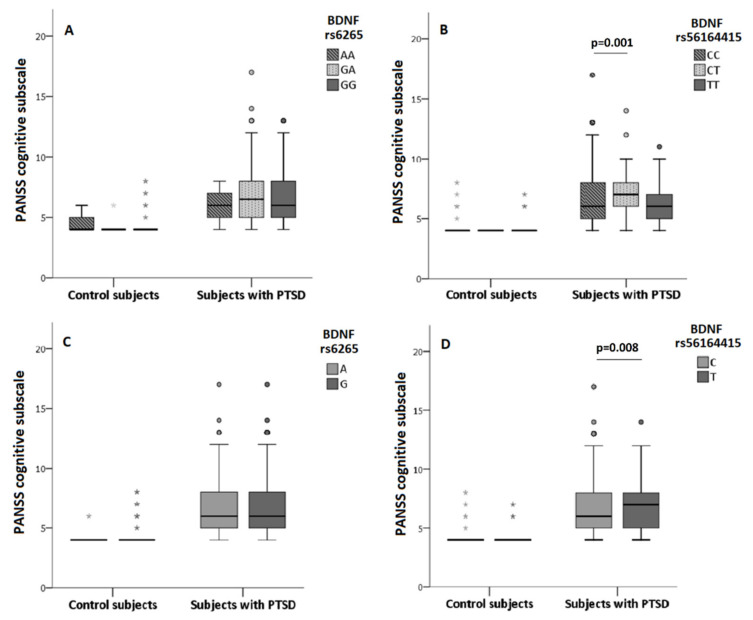
PANSS cognitive subscale scores in control subjects and subjects with PTSD depending on BDNF rs6265 (**A**,**C**) and BDNF rs56164415 (**B**,**D**) genotypes and alleles. Results are shown using the box and whiskers plot, where the central box represents the interquartile range (IQR), middle line the median, whiskers 1.5 IQR, dots the outliers and grey asterix extreme values. The differences between groups were analyzed using Kruskal–Wallis ANOVA, followed by the Dunn’s test, or Mann–Whitney U test.

**Figure 2 biomolecules-11-00641-f002:**
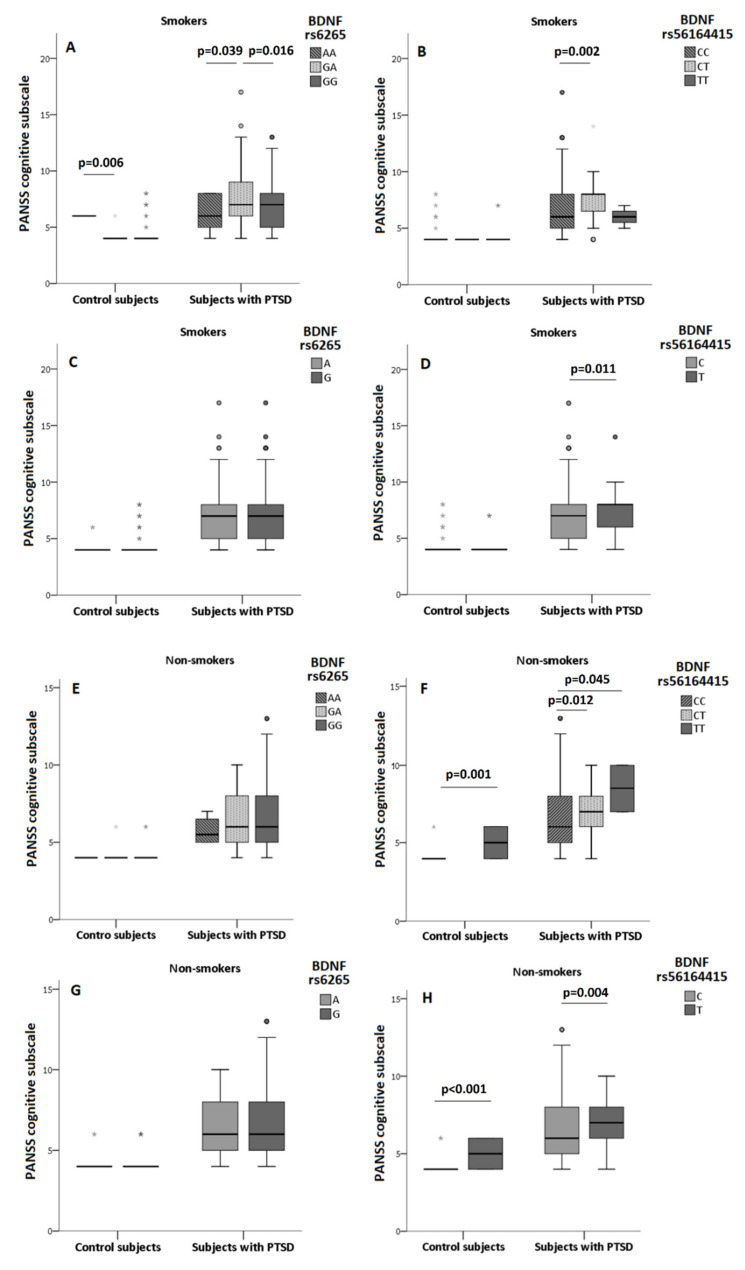
PANSS cognitive subscale scores in control subjects and subjects with PTSD, divided by smoking status, depending on BDNF rs6265 (**A**,**C**,**E**,**G**) and BDNF rs56164415 (**B**,**D**,**F**,**H**) polymorphisms. Results are shown using the box and whiskers plot, where the central box represents the interquartile range (IQR), middle line the median, whiskers 1.5 IQR, dots the outliers, and grey asterix presents extreme values. The differences between groups were analyzed using Kruskal–Wallis ANOVA, followed by the Dunn’s test, or Mann–Whitney U test.

**Table 1 biomolecules-11-00641-t001:** Demographic and clinical data of 121 control subjects and 643 subjects with PTSD. The data are shown as the median and interquartile range (25th; 75th percentile) or as total number N (frequency in percentages). *p* values in bold represent statistical significance.

		Control Subjects	Subjects with PTSD	Statistics
Age/Median (25th; 75th)		59 (52; 67)	50 (41; 56)	U = 21,941.0; ***p* < 0.001**
BMI/Median (25th; 75th)		29.1 (26.4; 30.5)	27.8 (25.7; 30.4)	U = 16,725.0; *p* = 0.053
CAPS/Median (25th; 75th)		-	81 (72; 90)	-
CAPS severity/N (%)	Mild	-	94 (14.9)	
	Moderate	-	443 (68.9)	
	Severe	-	106 (16.5)	
PANSS scores		32 (30; 33)	54 (48; 61)	U = 1720.5; *p* < **0.001**
ROCF copy		20 (20; 20)	20 (20; 20)	U = 11,905.0; *p* = 0.128
ROCF immediate recall		19 (18; 20)	12 (9; 15)	U = 1684.0; ***p* < 0.001**
ROCF delayed recall		17 (16; 18)	7 (5; 9)	U = 1598.0; ***p* < 0.001**
Smoking/N (%)	Yes	67 (55.8)	344 (67.2)	χ^2^ = 4.636; df = 1; *p* = 0.031
	No	53 (44.2)	175 (34.2)	
Number of cigarettes/Median (25th; 75th)		10 (10; 16)	18 (15; 22)	U = 3723.0; *p* < **0.001**

CAPS—Clinician Administered PTSD Scale; PANSS—Positive and Negative Syndrome Scale; U = Mann–Whitney test statistics value; ROCF—Rey-Osterrieth Complex Figure test; PANSS and ROCF scores are presented as Median (25th; 75th); χ^2^ = chi-square test.

**Table 2 biomolecules-11-00641-t002:** The frequencies of the BDNF rs6265 and BDNF rs56164415 genotypes and alleles in 643 veterans with PTSD and 121 control subjects. The data is shown as total number N (frequency in percentages). *p* values in bold represent statistical significance.

SNP	Genotype/Allele	Control Subjects	Subjects with PTSD	Total	Statistics
BDNF rs6265	AA	4 (3.3)	18 (2.8)	22 (2.9)	χ^2^ = 1.379; df = 2; *p* = 0.502
	GA	48 (39.7)	222 (34.5)	270 (35.3)	
	GG	69 (57.0)	403 (62.7)	472 (61.8)	
	A carriers	52 (43.0)	240 (37.3)	292 (38.2)	χ^2^ = 1.377; df = 1; *p* = 0.241
	GG	69 (57.0)	403 (62.7)	472 (61.8)	
	A	56 (23.1)	258 (20.1)	314 (20.5)	χ^2^ = 1.182; df = 1; *p* = 0.277
	G	186 (76.9)	1028 (79.9)	1214 (79.5)	
BDNF rs56164415	CC	105 (86.8)	473 (73.6)	578 (75.6)	χ^2^ = 58.335; df = 2; ***p* < 0.001**
	CT	2 (1.7)	159 (24.7)	161 (21.1)	
	TT	14 (11.6)	11 (1.7)	25 (3.3)	
	T carriers	16 (13.3)	170 (26.4)	186 (24.4)	χ^2^ = 9.653; df = 1; ***p* = 0.002**
	CC	105 (86.8)	473 (73.6)	578 (75.6)	
	C	212 (87.6)	1105 (85.9)	1317 (86.2)	χ^2^ = 0.482; df = 1; *p* = 0.488
	T	30 (12.4)	181 (14.1)	211 (13.8)	

BDNF—brain derived neurotrophic factor; χ^2^ = chi-square test.

**Table 3 biomolecules-11-00641-t003:** Scores of the PANSS total and PANSS cognitive subscales and the ROCF copy, immediate recall, and delayed recall tests in control subjects and veterans with PTSD, depending on smoking status. The data is shown as median and interquartile range (25th; 75th percentile), while *p* values in bold represent statistical significance.

	Control Subjects		Subjects with PTSD	
	Smokers	Non-Smokers	Smokers	Non-Smokers
PANSS total scores	32 (30; 34)	32 (30; 33)	57 (50; 62)	54 (48; 61)
	U = 1546.5; *p* = 0.210		U = 25,862.0; ***p* = 0.009**	
PANSS cognitive subscale scores	4 (4; 4)	4 (4; 4)	7 (5; 8)	6 (5; 8)
	U = 1691.5; *p* = 0.394		U = 26,315.0; ***p* = 0.018**	
ROCF Copy scores	20 (20; 20)	20 (20; 20)	20 (20; 20)	20 (20; 20)
	U = 308.0; *p* = 0.149		U = 1.019.0; *p* = 0.844	
ROCF Immediate Recall scores	19 (18; 19)	19 (18; 20)	12 (9; 15)	13 (11; 15)
	U = 305.5; *p* = 0.529		U = 956.0; *p* = 0.548	
ROCF Delayed Recall scores	17 (17; 17)	17 (16; 18)	7 (5; 10)	6 (5; 9)
	U = 282.0; *p* = 0.290		U = 939.5; *p* = 0.464	

PANSS—Positive and Negative Syndrome Scale; ROCF—Rey-Osterrieth Complex Figure (ROCF); U—Mann–Whitney test statistics value.

## Data Availability

The data presented in this study are available on request from the corresponding author. The data are not publicly available due to privacy and ethical restrictions.

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
