# Peer review of "Moderating Effects of BDNF Genetic Variants and Smoking on Cognition in PTSD Veterans"

_biomolecules, 2021, doi:10.3390/biom11050641_

Round 1

Reviewer 1 Report

The study investigates 2 groups of PTSD and controls for the effect of SNPs on tobacco use and psychological symptoms. It is a relevant question.   It is original enough for publication after the recommended minor changes.  The paper is well-written. The text is clear and easy to read.  The conclusions are consistent with the evidence and arguments presented.   The paper appropriately addresses the main question posed by the paper. 

The only comment I have are:

1- Delete the total n and % from all tables.

2- Add * for p value for all the Figures.

Author Response

The study investigates 2 groups of PTSD and controls for the effect of SNPs on tobacco use and psychological symptoms. It is a relevant question.   It is original enough for publication after the recommended minor changes.  The paper is well-written. The text is clear and easy to read.  The conclusions are consistent with the evidence and arguments presented.   The paper appropriately addresses the main question posed by the paper. 

The only comment I have are:

1- Delete the total n and % from all tables.

2- Add * for p value for all the Figures.

Answer: We have accepted this comment and deleted total N and % from Tables and added p values in the Figures.  

Reviewer 2 Report

The paper from Erjavec et al. is well written in a standard English, it is accurate, and the statistical analysis detailed as required.

However, there are some issues the authors should consider.

1.The main bias here is that the mean age of healthy control group is significantly higher (p<0.001) than PTSD veterans, although authors stated: “our results were controlled for the effects of age” (lines 400-401).

Indeed, there are several reports pinpointing the effect of younger age to develop PTSD, for several reasons, included that healthier people who survive into older ages are less likely to have current or lifetime PTSD. In fact, older age is associated with better emotion regulation, emotional well-being, and positivity biases in memory and appraisal (Charles, 2010; Charles & Carstensen, 2010). More importantly, further investigations are needed to understand how age-related gains intersect with PTSD or psychopathology. This could be discussed for a deeper understanding of differences found.

Cohort differences are known to underlie age group differences in symptom endorsement (minimization of symptoms due to shame or stigma by younger cohorts) and psychological mindedness.

-Age is a factor for smoking prevalence, and the higher mean age of controls may mask a possible difference (in fact, a nominal difference is found). This is not considered in discussing the data presented.

-Is it possible to re-evaluate the data by using e.g., only 45-55 years individuals from both groups, to avoid unwanted age-dependent bias?

- PTSD has also been linked to metabolic conditions. Experience of PTSD symptoms is associated with an increased risk of becoming overweight or obese, and PTSD symptom onset as well as ageing alter BMI trajectories over time. How this affect the analysis?

  1. It is not clear why non-PTSD veterans -allowing for discrimination of resilience-dependent factors- were not included in the study.

Reviewer 3 Report

In the present study, the Authors aimed to evaluate changes in cognitive function in war veterans with PTSD and control subjects, controlled for the effect of smoking, and hypothesized that the A allele of the BDNF rs6265 and T allele of the BDNF rs56164415 were associated with cognitive decline in war veterans with PTSD.

Overall, I found the paper timely, well written, very interesting and scientifically sound. It adds something to the relatively poor literature concerning the cognition in subjects with PTSD. I have only some minor suggestions aimed to improve the high quality of the paper and these are outlined below:

1) In the introduction, it should be mentioned that one of the risk factors of PTSD might be alexithymia, that may on its own influence cognition, with appropriate references (see doi: 10.1080/13651501.2019.1699575).

2) As well for a normal BDNF release to prevent or treat PTSD it seems important the correct functioning of noradrenergic system. Please add a brief note on this point wirh appropriate reference (see doi: 10.2174/1389450116666150506114108).

3) The study included 643 male war veterans with combat related PTSD and this is a huge sample. But how many subjects were screened and refused to participate? As well were subjects consecutive or randomly selected?

4) As drug abuse or alcohol dependence within three months prior to admission was correctly an exclusion criterion, I guess how many subjects with these problems were excluded as we know that SUD is higlhly prevalent in PTSD.

5) How intellectual disability was assessed? Please specify.

6) I guess if the PTSD duration of illness may influence results in a significant way. Please add more informations on this point.

7) It's unclear wheter the PTSD subjects were taking medications for the disorders. This should be specified and commented.

8) In some point of the paper the English language needs to be carefully checked by a native speaker.
